# Global increase in tropical cyclone rain rate

Oscar Guzman ⬤ [1] & Haiyan Jiang[1✉]

Theoretical models of the potential intensity of tropical cyclones (TCs) suggest that TC rainfall rates should increase in a warmer environment but limited observational evidence has been studied to test these hypotheses on a global scale. The present study explores the general trends of TC rainfall rates based on a 19-year (1998–2016) time series of continuous observational data collected by the Tropical Rainfall Measuring Mission and the Global Precipitation Measurement mission. Overall, observations exhibit an increasing trend in the average TC rainfall rate of about 1.3% per year, a fact that is contributed mainly by the combined effect of the reduction in the inner-core rainfall rate with the increase in rainfall rate on the rainband region. We found that the increasing trend is more pronounced in the Northwestern Pacific and North Atlantic than in other global basins, and it is relatively uniform for all TC intensities. Further analysis shows that these trends are associated with increases in sea surface temperature and total precipitable water in the TC environment.

[1] Department of Earth & Environment, Florida International University, Miami, FL, USA. ✉email: haiyan.jiang@fiu.edu

Over the last decades, precipitation associated with tropical cyclones (TCs) has received more and more attention from the scientific community because of its potential to trigger freshwater flooding and landslide events that have caused the loss of lives and immeasurable damages in coastal areas[1–4]. Current theoretical models suggest that TC potential intensity is expected to increase with the increasing global mean sea surface temperatures[5,6].

It has been reported that TC's destructive power has increased during the past few decades[7], but it is still under debate whether the number and intensity of TCs have increased under global warming due to limitations in data availability of long-term homogenous TC records[8–11]. Nevertheless, numerical models have predicted that the increase in rainfall rates associated with TCs is an expected response of increased evaporation from the ocean surface and increased moisture in the atmosphere on global warming scenarios[12–16]. In the most recent Intergovernmental Panel on Climate Change (IPCC) report, Christensen et al.[17] noted that an important majority of the models predict an increasing range between 5 and 20% with consistent results in all TC basins. That finding has received more support from the scientific community, as noted by the Knutson et al.[18] in Part II of the Tropical Cyclone and Climate Change Assessment, in which they combined rainfall projections from multiple model-based studies to create an aggregate result of 14% TC rain rate increase for a 2 °C temperature increase, or broadly close to the rate of tropical water vapor increase expected for warming at constant relative humidity. Knutson et al.[18] also suggested a lower confidence level of the TC rainfall rate increase for Southern Hemisphere basins.

However, the availability of continuous observations during long periods has limited the validation of these numerical predictions over TC-prone basins at a global scale. Although previous research has not yet detected any global trends in TC rainfall rates[19], Lau and Zhou[20] found an increasing trend of total TC lifetime accumulate rain in the Atlantic and a decreasing trend in the northeast and northwest Pacific during two recent decades (1988–2007) using relatively low-resolution Global Precipitation Climatology Project (GPCP) data. They recognized that the total TC lifetime accumulated rain measure derived from the low-resolution rainfall data is not ideal and determining realistic trends of TC rain rates will require long-term (multi-decadal), high temporal (3-hourly or at least daily), and spatial resolution (25 km or less) rainfall data[20].

In this study, we examine how the TC rainfall rate has changed in all TC-prone basins, including Northern Atlantic (ATL), East-Central Pacific (ECPA), Northwestern Pacific (NWP), Northern Indian Ocean (NIO), Southern Indian Ocean (SIO), and southern Pacific (SPA); using the 3-hourly, 0.25 × 0.25 degrees of the Tropical Rainfall Measuring Mission (TRMM) 3B42 data for almost two decades (1998–2016) derived from TRMM and Global Precipitation Measurement (GPM) missions. Globally, we found an increasing trend that is more pronounced in the northern hemisphere basins (except for the ECPA). Analysis of environmental parameters shows that these trends are associated with increases in sea surface temperature and total precipitable water that can affect the TC rainband environment. In contrast with the northern hemisphere, southern hemisphere basins show much smaller increasing trends due to slight variations of the environmental conditions in that part of the world. These results have important implications in understanding TC rainfall mechanisms and evaluating current climate models for future projections under a warming environment.

## Results

**Global trends within the total TC rainfall area.** Conventionally, the total TC rainfall area is defined by the 500 km radius threshold. This approach was used by several researchers in the previous studies[21–24]. The rationale behind this distance threshold is explained by the findings of Englehart and Douglas[25], which indicates that in 90% of the TC cases, the distance between the storm center and the outer edge of its cloud shield is <550–600 km. However, we realize that the size of TCs can vary substantially across different basins, individual storms, and even within the lifetime of the same storm[26]. As suggested by Stansfield et al.[27], a hard-coded radius of 500 km could overestimate TC-related overland precipitation. Therefore, in this study, we define TC total raining area using a framework based on the concept of tropical cyclone precipitation features (TCPF)[22]. In this TCPF framework, a precipitation feature (PF, or a rain cell) is defined by grouping contiguous pixels based on certain criteria. In this case, the criterion is a 3B42 rain rate >0.1 mm/h (see section "TC rainfall determination" for more details). To be qualified as a TCPF, the distance between the TC center and the geometric center of the PF must be <500 km. We compared the traditional simple truncation of 500 km radius from the TC center (Fig. 1a) and the TCPF-based TC total rain area definition (Fig. 1b). Although both methods exhibit similar and consistent results, our findings focus on the TCPF approach since this provides a better estimation of the TC size[22].

Global TC rainfall trends are determined using two statistical methods, linear regression fittings, and Mann–Kendall trend tests (see "Methods" section). Overall, results exhibit an increasing trend in the hourly precipitation rate of 0.027 mm h$^{-1}$ year$^{-1}$, with a coefficient of determination $R^2$ of 0.90 (Fig. 1b). This trend can be interpreted as an average percentage of change near +1.3% from one year to the next during the study period (Supplementary Table 1). The result of Kendall's test demonstrates the presence of a trend with 99.9% confidence, with a strong relationship of the TC rainfall rate with respect to ordinal variations of time (Kendall's tau = 0.80). Additional validation of the trend is performed with calculations of Sen's slopes, which ratifies the result of the linear regression and suggests 0.022 and 0.031 mm h$^{-1}$ year$^{-1}$ as the maximum upper and lower bounds of the slope with 95% confidence, respectively (Supplementary Table 2).

Inter-basin comparisons show that this increasing trend is more substantial in the Northwestern Pacific and Northern Atlantic than in other basins, as represented by increased hourly precipitation rates of as much as 0.04 mm h$^{-1}$ year$^{-1}$. Their determination coefficients, Kendall's tau, and statistical significance are also more robust (Fig. 2a, c). Rising trends across the Northern Indian Ocean are centered around 0.03 mm h$^{-1}$ year$^{-1}$ (Fig. 2d), and trends in the East and Central Pacific are near 0.018 mm h$^{-1}$ year$^{-1}$ (Fig. 2b). In contrast with the northern hemisphere basins (excepting ECPA), the Southern Pacific and Southern Indian Ocean exhibit a minor trend, as demonstrated by their shallow slopes (Fig. 2e, f) and slightly lower statistical significances. A summary of the main inter-basin statistics is presented in Supplementary Table 3.

With reference to the TC intensities, Supplementary Fig. 1 and Supplementary Table 4 summarize the trends and their significance level at a global scale. Since the denomination of intensity categories varies depending on the regional ocean basins where TCs form, in this study, we define tropical depression (TD) as a system with wind speed from 10 to 33 knots, tropical storms (TS) as systems with wind between 34 and 63 knots, and the hurricane categories 1–5 (CAT1, CAT2, CAT3, CAT4, and CAT5) are adopted from the Saffir-Simpson scale. In all the categories, there is a consistent increase in the range of 0.015–0.043 mm h$^{-1}$ year$^{-1}$. The most pronounced increases occur in the extremes of the storm category scale (TD, TS, CAT4, and CAT5) with a slightly more pronounced effect in the CAT4 and CAT5 hurricanes with slopes of 0.03 and 0.04 mm h$^{-1}$ year$^{-1}$, respectively (Supplementary Fig. 1a, b, f, g).

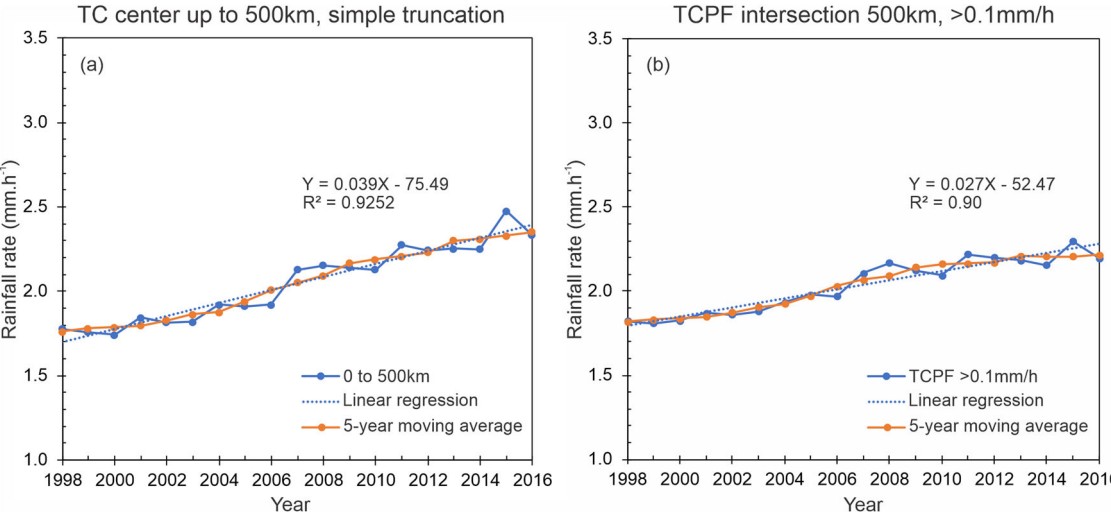

**Fig. 1 Time series and the linear regression fit of TC rain averaged within the total TC rainfall area for TCs in all global TC-prone basins during 1998-2016. The total TC rainfall area is defined by. a** Simple truncation method, and **b** tropical cyclone precipitating feature method. The linear fitting function and square of the correlation are indicated. The 5-year moving average is also shown.

Intermediate storm categories show smaller increases (Supplementary Fig. 1c, d, e), specially CAT2 hurricanes with a slope of only $0.015\,\mathrm{mm\,h^{-1}\,year^{-1}}$.

**Differentiated trends: inner-core vs. rainbands**. In the case of the inner-core region, we used a storm-dependent method that uses the radius of maximum azimuthal rain rate (RMR) as the size determining factor. A sensitive test comparing a simple truncation of 100 km, RMR, 1.5×RMR, and 2×RMR was performed to describe the effects of varying the selection distances in estimating averaged inner-core rainfall rates (Supplementary Fig. 2). Results indicate that the slopes of averaged rainfall rates slightly decrease as a longer radius is used to define the inner-core region. However, in all the inner-core definitions, the nearly same result is obtained. Therefore, we adopted a distance threshold of 2*RMR to ensure the full inclusion of the most developed cumulus nimbus, area of high winds, intense vertical motions, and the eye, as suggested by Shea and Gray[28]. Finally, for the rainband region, we used the area from 2×RMR until the external borders defined by the TCPF method that includes precipitation features with centroids within 500 km from the TC center; this selection was made as an approximation to allow the inclusion of rainfall rates in the area beyond two times the radius of maximum wind and the outermost part of the storm.

In the inner-core and rainband regions, results indicate that, while the rainfall rate in the inner-core decreases, the rainfall rate in the rainband region increases (Fig. 3a, b). This inverse relationship is present in all intensity categories; however, in the inner-core region, the Kendall test shows that it is only statistically significant for storms from CAT1 to CAT5 (Supplementary Table 2). In both storm regions, linear regressions reach high R-squared values, particularly for the rainband where the trend is more homogenous. In a more detailed analysis, both time series seem to follow differentiated slopes for the periods before and after 2003–2004, suggesting a potential inflection point (or peak in the inner-core) that could be associated with decadal or multi-decadal cycles.

## Discussion

In this study, an increasing trend in the TC rainfall rate during the period 1998–2016 has been identified on a global scale. On average, an increase of $0.027\,\mathrm{mm\,h^{-1}\,year^{-1}}$ is observed in the time

series. This trend is nearly equivalent to a percentage of change of +1.3% from one year to the next. The increase is controlled by the reduction of the averaged inner-core rainfall in a range between $-0.011$ and $-0.096\,\mathrm{mm\,h^{-1}\,year^{-1}}$ (percentage of change near to $-1.5\%/\mathrm{year}$, for CAT1 to CAT5) along with an increase in the averaged rainband rainfall rate of $0.035\,\mathrm{mm\,h^{-1}\,year^{-1}}$ (percentage of change near to $+1.4\%/\mathrm{year}$). Despite the fact that the inner-core region produces the most intense rainfall rates (Supplementary Fig. 3a, b), the extent of the rainband region is much larger than the inner-core, which leads to a more significant contribution from the rainbands to the total TC rain (Supplementary Fig. 3c, d). These differences in the proportional contribution and the rising trend in the rainbands suggest that the global increase must be mainly a consequence of special conditions favoring rainfall production in the rainband environment.

We hypothesize that the increasing global trend is associated with higher availability of water vapor triggered by warmer sea surface temperatures under constant relative humidity conditions, as suggested by the calculations of Knutson et al.[18] using the Clausius–Clapeyron equation, as well as the projections of tropospheric water vapor in the IPCC report[29]. To check this hypothesis, we analyzed the Reynolds sea surface temperature (SST)[30] and the total precipitable water (TPW) at $t = 0$ from the Global Forecasting System (GFS) analysis in the region from 0 to 500 kilometers, both variables as reported in the Statistical Hurricane Intensity Prediction Scheme (SHIPS) developmental database[31–34]. Results confirm that at least in the northern hemisphere (NH), the environment has been warming and moistening during the study period (Fig. 4a, c), which seems to favor the precipitation rates in the rainband region. Besides, linear regressions of both SST and TPW exhibit robust statistical significance and confirmation of a rising trend from the results of the Kendall test (Supplementary Table 5). We also found positive correlations between precipitation rate and SST with a correlation coefficient of 0.22, and between precipitation rate and TPW with a correlation coefficient of 0.50.

In contrast to the northern hemisphere findings, our results over the southern hemisphere basins show no significant trend in the SST and TPW values (Supplementary Table 5). This situation is explained by the smaller slope of the sea surface temperature when compared to the northern hemisphere average (Fig. 4a, b). Here, although the linear regression of sea surface temperature shows a slight decreasing slope, the Kendall test indicates this

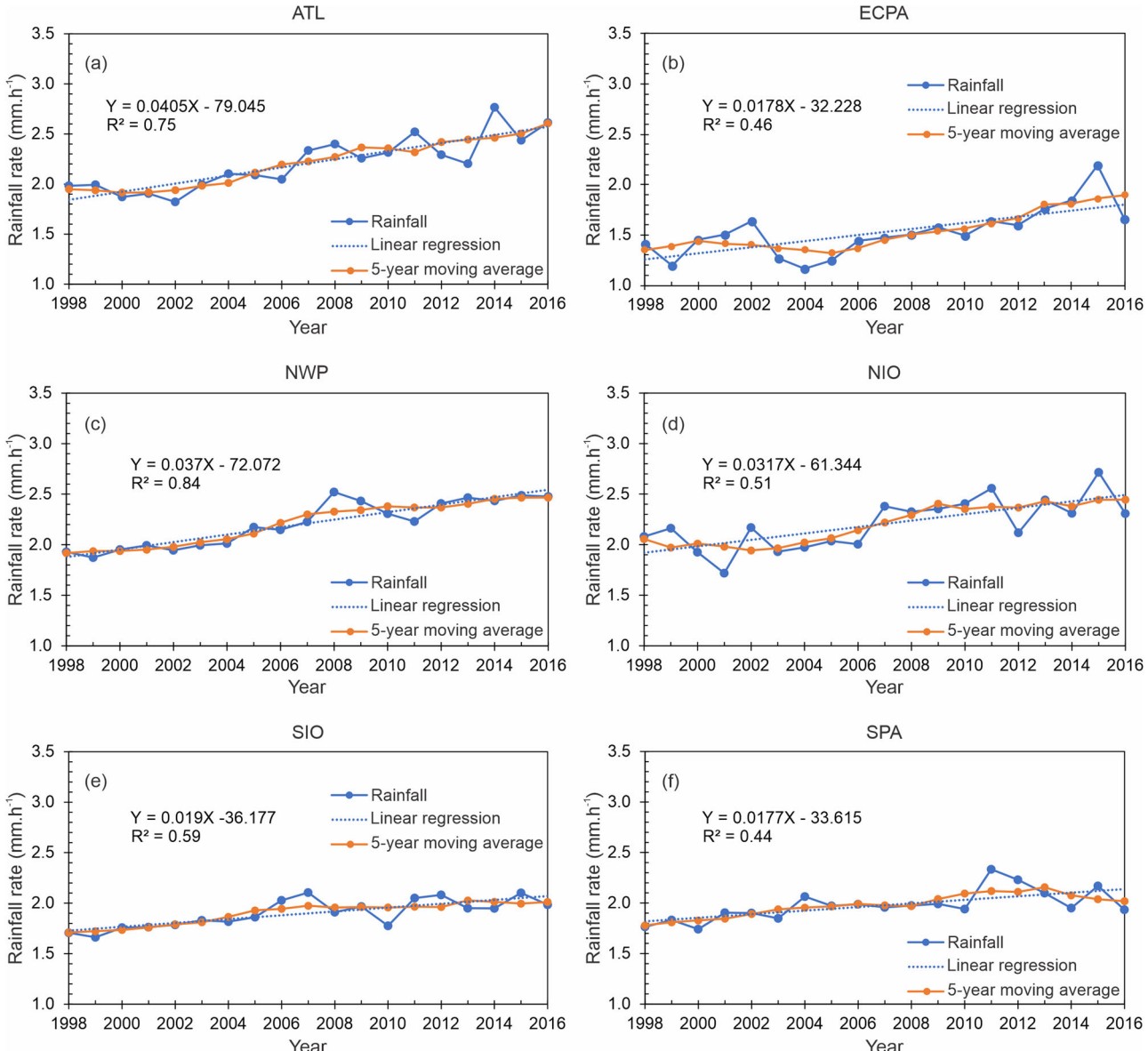

**Fig. 2 Time series and the linear regression fit of TC rain averaged within the total TC rainfall area defined by the tropical cyclone precipitating feature method for TCs in 6 different basins. a** Northern Atlantic, **b** East-Central Pacific, **c** Northwestern Pacific, **d** Northern Indian Ocean, **e** Southern Indian Ocean, and **f** southern Pacific. The linear fitting function and square of the correlation are indicated. The 5-year moving average is also shown.

trend approaches zero (Supplementary Table 5). A similar situation occurs with the availability of total precipitable water (Fig. 4c, d); while the northern hemisphere is increasing, the trend in SPA remains stable (Supplementary Table 5). Figure 4d exhibits a slightly negative slope, but the Kendall test indicates it is still approaching zero from the statistical perspective. The combined results of the sea surface temperature and total precipitable water are consistent with our observational findings on the occurrence of differential rainfall rate increases between both hemispheres, which suggest slightly less pronounced effects of the warming projections in the increase in rainfall rates along the southern hemisphere. This is consistent with the climate modeling results summarized by Knutson et al.[18].

As a potential and desired alternative explanation, we also explored the change of maximum sustained wind speed during the analysis period (Supplementary Table 5). However, statistical results show that storm intensities remain unchanged; therefore, we concluded that the increasing global trend must be the result

of higher SST and TPW only. In other words, the scheme that assumes that the magnitude of rainfall is totally dependent on storm intensity is less than perfect because of the influence of environmental conditions.

We found that the results of Kim et al.[35] may partially support our findings. Their study suggests that the strength of inner-core rainfall strength is strongly correlated with TC intensity, but weakly correlated with large-scale environmental conditions, while the TC total rainfall area showed a stronger correlation with environmental conditions than with TC intensity. The total TC raining area is closely related to the rainband environment. In other words, TC rainfall outside the core is more sensitive to environmental conditions than that inside the core, which may explain the finding that only rainfall in the rainband increased with increasing SST and TPW.

This study uses a method that employs the radius of maximum azimuthal rain rate (RMR) to define the inner-core region. Traditionally, the radius of maximum wind (RMW) has been a more

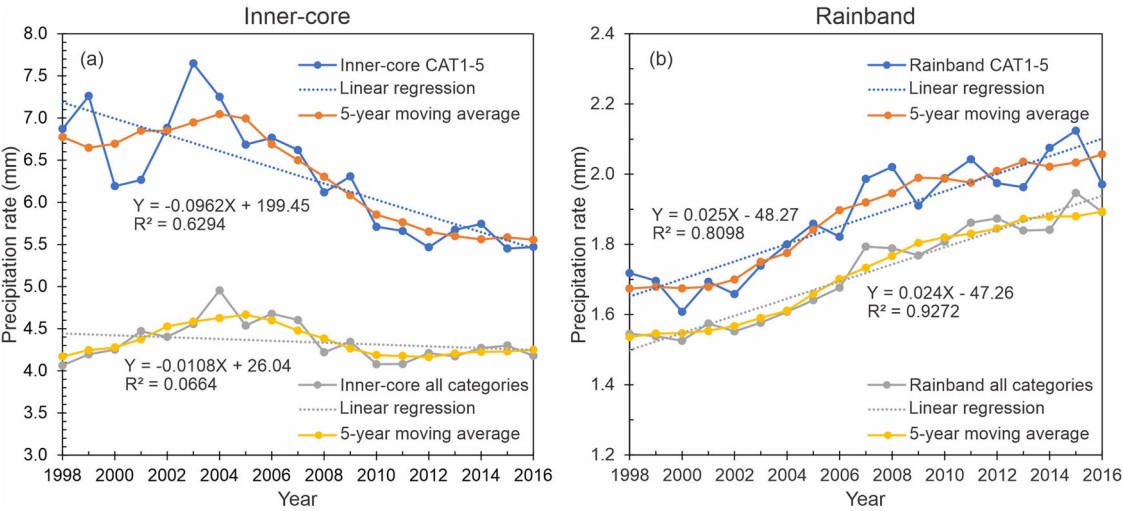

**Fig. 3 Time series and the linear regression fit of averaged rain for different TC regions. a** mean TC inner-core rain, and **b** mean TC rainband rain in all global TC-prone basins. Plots are showing averages for all intensity categories and for hurricanes from CAT1 to CAT5. The linear fitting function and square of the correlation are indicated. The 5-year moving average is also shown.

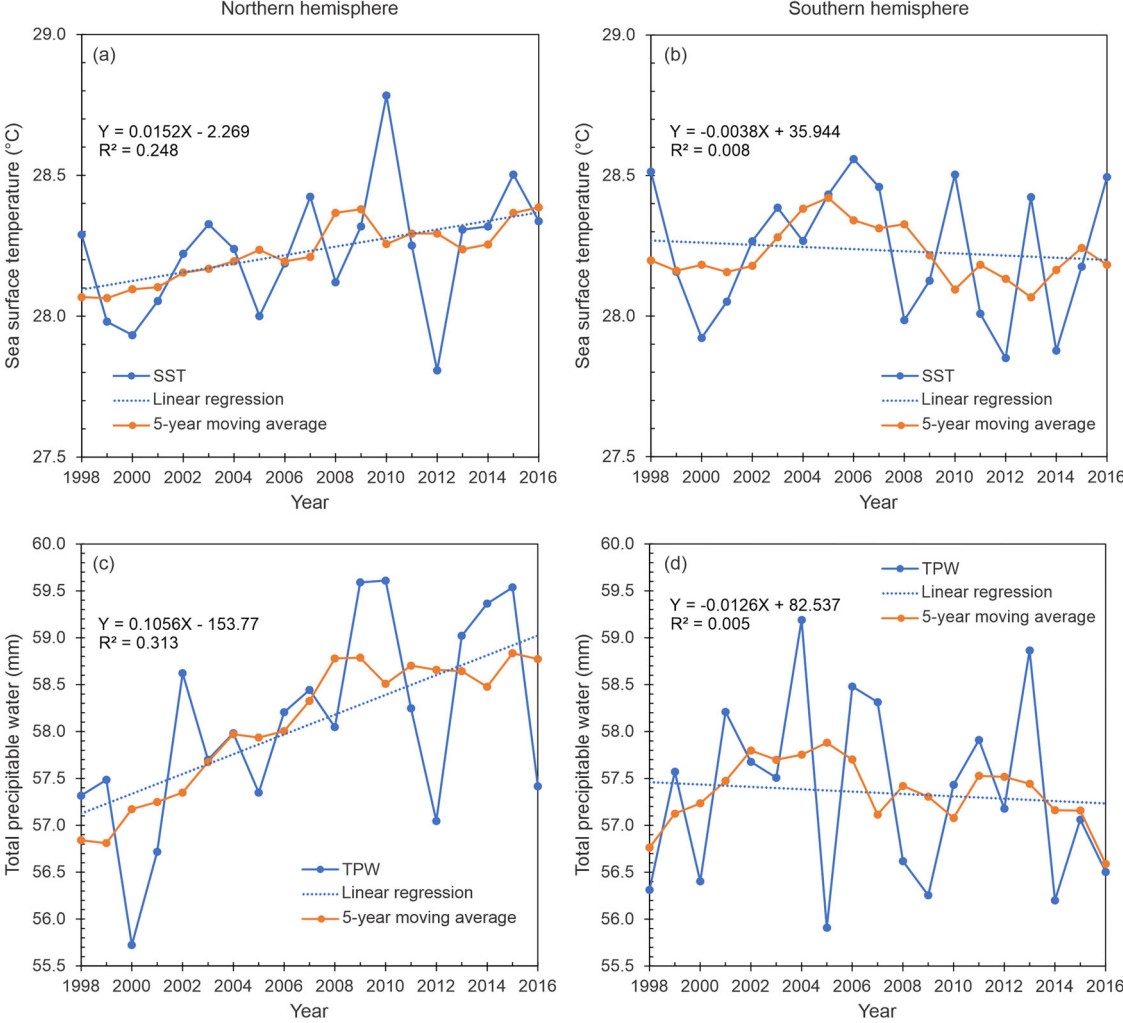

**Fig. 4 Time series and the linear regression fit of TC environmental parameters from 0 to 500 km. a** Sea surface temperature in the northern hemisphere, **b** sea surface temperature in the southern hemisphere, **c** total precipitable water in the northern hemisphere, and **d** total precipitable water from in the southern hemisphere. The linear fitting function and square of the correlation are indicated. The 5-year moving average is also shown.

conventional parameter to do so. However, RMW data is not available for all global TCs analyzed in this research, driving us to adopt the RMR as a proxy of RMW. Considering that one of our observational results indicates that the inner-core rainfall rate is decreasing, further analysis of the RMR was performed. We found that globally and when considering all the TC categories, the RMR does not observe any particular trend (Supplementary Fig. 4a–c). However, in CAT1 to CAT5 storms, results show that the RMR slightly expands outwards (Supplementary Fig. 4d–f) with lower $R^2$ and confidence levels than other reported trends (Supplementary Table 2).

Overall, our observational results coincide with the modeling approaches that suggest increasing trends in TCs rainfall rates with warmer sea surface temperatures[12–16]. However, the differences in the order of magnitude between the expected (modeled) vs. observed rainfall rates in our period of study are substantial; while modeling averages projects a +14% for a 2 °C warming increase[18], our result for the 19-year time series is close to +21% for a variation of 0.21 °C only (with more presence in the northern hemisphere). We consider this difference is explained by the domaining influence of the increasing rainfall rates in the rainband region, which is mainly attributed to the more availability of precipitable water in the atmosphere.

In addition to the difference mentioned above, we found that rainfall rates at the inner-core level decrease for CAT1 to CAT5 hurricanes, which contradicts the results of modeling projections that suggest potential increases within 100 km of the TC center[18]. However, this finding is actually similar to the results of a recent observational study by Kim[36], which suggested that the inner-core rainfall rate decreases by 0.081 mm h$^{-1}$ per year (his Fig. 3.1.6). Kim[36] used the same TRMM 3B42 data, but the analysis period was different and tropical depressions were excluded. The definition of inner-core rainfall was also different from this study. He used a fixed 100 km radius from the TC center to define inner-core rainfall. So far, we do not have a good explanation why the rain rate in the inner-core has a decreasing trend. This would be a direction for future studies.

An additional potential contradiction with our results is the findings of Lau and Zhou[20]. They reported differences in the sign and magnitude of total TC lifetime accumulate rain trends between ATL (+23% per decade), ECPA (−25.1% per decade), and NWP (−20.9% per decade), which seemingly agrees with our findings in ATL but amply differed for ECPA and NWP. However, as mentioned in the introduction, Lau and Zhou (2012)[20] used the GPCP data, which is a 5-day mean rainfall dataset from satellite and rain gauge measurements with a spatial resolution of 2.5° latitude X 2.5° longitude. With such a low temporal resolution, it was impossible to estimate realistic rain rate intensity in TCs. They recognized that the total TC lifetime accumulate rain measure they used was rather an integral measure of total rain energy associated with a TC. Therefore, their results are not comparable with our study here.

The above analyses lead us to conclude that the increasing rainfall rate induced by warmer and moister environments has the potential to generate more severe impacts associated with TC rainfall events across TC-prone areas, particularly in the Northwestern Pacific and Northern Atlantic basins where the increasing trends are more pronounced.

## Methods

**Data**. The time series cover the period from 1998 to 2016 including all the basins with TC activity. A total of 81,242 satellite observations are analyzed with the intensity categories and per-basin sample distribution shown in Supplementary Table 6. Yearly samples oscillate between 3171 and 5315, with an average of 4232 observations per year (Supplementary Table 7). TCs in the Southern Atlantic basin are excluded because of their unusual occurrence and insufficient sample size.

Precipitation information is extracted from TRMM 3B42 satellite estimations (version 7)[37] for the period from January 1998 to September 2014, and the transitional product that uses the new GPM constellation for the period from September 2014 to December 2016. This product was selected as the most extended available time series with homogeneous spatial-temporal data over tropical areas (3-hourly with 0.25° × 0.25° resolution), and it was used to characterize TC precipitation with the best possible climatological significance.

TC characteristics, including location, time, and intensity, are extracted from the International Best Track Archive for Climate Stewardship (IBTrACS version 4)[38]; this database combines best-track records from different meteorological authorities on a global basis. Considering that some relevant parameters might present differences between reporting agencies, we privileged the information from the US agencies, particularly from the Joint Typhoon Warning Center (JTWC) for the TCs occurring in the Pacific Ocean Basins. Only storms in which the 3B42 extent covers the entire TC size area are considered. To meet this criterion, those best-track data points with TC centers located beyond 44°N and 44°S (about 660 km before the edge of the 3B42 border) are removed.

Sea surface temperature and total precipitable water estimations used to characterize the environmental conditions around the storms are extracted from the Statistical Hurricane Intensity Prediction Scheme (SHIPS, version available in July 2018). This developmental database includes estimations of a wide variety of environmental parameters over various annular regions calculated from the TC center[31–34]. For the current study, only the values for $t = 0$ are employed in our analyses. Reynolds sea surface temperature (RSST) and total precipitable water (TPW) are averaged between 0 and 500 km from the storm center.

**TC rainfall determination**. The algorithm begins selecting the rain cells within the searching range from the 3B42 precipitation data with a spatiotemporal match to every 3-h position of the best-track data. Storms in the vicinity 50°N or 50°S are only included when they fully comply with the areal coverage criteria established by the searching range. As mentioned previously, in this study, we use storm-dependent definitions to define the total TC raining area and the inner-core raining area. In the TCPF framework[22], to define the total TC raining area, a minimum value of precipitation rate must be used to group each PF or rain cell. Ideally, this threshold should be rain rate = 0 mm h$^{-1}$. However, the infrared or microwave rainfall retrievals in TRMM/GPM 3B42 are often contaminated by non-raining signals, especially for light rains in the retrievals. This contamination could cause problems in our TCPF identification. Therefore, we tested a few minimum values as the threshold to define rain cells, including 0.01, 0.1, and 0.25 mm h$^{-1}$. The sensitivity test confirms that the rising global trend still presents in all scenarios. The intermediate value of 0.1 mm h$^{-1}$ is chosen in this study. The average rainfall rates are calculated by considering the rain cells showing a precipitation rate over 0.1 mm h$^{-1}$. Zero values and non-data values are excluded from the calculations. Once each average is obtained for the corresponding TC region, a yearly average is calculated, and a set of time series is created for different intensity and different basin comparisons (Figs. 1–3, and Supplementary Fig. 1).

Additionally, an axisymmetric decomposition is performed to determine the radius of the maximum azimuthal rain rate (RMR) for each TRMM 3B42 TC observation. We find the RMRs by locating the rainfall rate maximum from the azimuthal mean values derived from the wavenumber zero of the Fourier transformation, as suggested in previous investigations[12,39]. This RMR definition is the same as in Shimada et al.[40] and Guzman and Jiang[41]. Then, a global time series with the yearly-averaged RMR values is created (Supplementary Fig. 4) and processed using the same approach used for the total TC rainfall, inner-core, and outer rainband regions.

**Time series**. The time series processing consists of simple linear regression analyses and statistical trend estimations using the Mann–Kendall test. For the linear regressions, time is used as the single predictor variable, and the corresponding regression equations and determination coefficients are calculated, including a verification of the normal distribution of the errors around zero. In the case of the Kendall test, the values of Kendall's tau and the $p$-values for the rejection or acceptance of the null hypothesis for different confidence levels are reported. Sen's slope with their corresponding bounds at 95% of confidence is also part of the calculations for alternative interpretation concerning the linear regression with the presence of potential outliers (Supplementary Tables 2–5).

The 5-year moving average was calculated in two steps: (1) Values from 2000 to 2014 result from the traditional method. For instance, the first averaged value is obtained by taking the average of a 5-values subset (2 before, current, and 2 after) shifting forward until the end of the time series. (2) In the case of 1998, 1999, 2015, and 2016, we use the moving average model (MA Smoothing Model) to estimate the most suitable values. In the process, we tested several $q$-values and determined that $q = 2$ is the best choice in all the cases. This value was selected by running the model over the entire series and then looking for the nearest series to the 5-year moving averages for 2000–2014. Finally, we included the modeled tails into the 5-year moving average to fully cover the period.

## Data availability

This paper uses satellite data openly available through the NASA Precipitation Processing System servers (3B42V7 product available at https://arthurhouhttps.pps.eosdis.nasa.gov/trmmdata/ and data from the open repository of the SHIPS developmental database from Colorado State University for all global TC-prone basins (5-day predictors available at https://registration.pps.eosdis.nasa.gov/registration/). DOIs and further detail on the data sources have been properly cited and provided in the main article and the References section. In addition, all data and analysis results used during the study are available on request from the authors. Please contact the corresponding author at haiyan.jiang@fiu.edu.

## Code availability

Code for data decoding, cleaning, and analysis associated with the current research was created using standard scientific programming packages. Copy of the scripts for python and IDL users is available from the corresponding author on reasonable request.

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

## Acknowledgements

This research is part of the first author's Ph.D. study. The first author was supported by the Fulbright-Colciencias grant as part of the cultural exchange program between the United States of America and the Republic of Colombia. As the PI, the second author would like to thank the following funding support: NSF Grant No. 1947304 under the direction of Dr. Jielun Sun, NASA Weather And Atmospheric Dynamics (WAAD) Grant NNX17AH72G under the direction of Drs. Ramesh Kakar and Gail Jackson, and NOAA Joint Hurricane Testbed (JHT) Grant NA17OAR4590142 under the direction of Mr. Richard Fulton.

## Author contributions

All the authors contributed to the central ideas presented in the paper, including data analysis and paper preparation. G.O. wrote the computer code, processed the data, and conducted the calculations. J.H. verified the analytical methods and supervised the findings of this work. All the authors discussed the result and contributed to the final manuscript.

## Competing interests

The authors declare no competing interests.
