## [Peer Review File · Nature Communications]

Reviewer comments, first round –

Reviewer #1 (Remarks to the Author):

The manuscript presents an observational evidence of global increase in tropical cyclone (TC) rainfall. There are many predictions that TC rainfall will increase due to global warming, but observational evidence to support this have been rare. This study is highly informative and well-organized, but there are several issues that need to be supplemented before publication.

1. One of major findings in this study is the reduction in the inner-core rainfall rate. However, it seems that this result can be changed according to the definition of inner-core rainfall. In general, the inner-core means the area containing the eyewall where the strongest precipitation occurs. During the analysis period, the radius of maximum rain (RMR) increased from 95 to 105 km approximately. Therefore, the 100 km radius seems to be rather small to define the inner-core area. It seems that more accurate changes in inner-core rainfall can be found if the 120 or 150 km radius is used instead of the 100 km radius. In addition, similar results can be found in Kim (2020) which suggested that the inner-core rainfall rate decreases by 0.081 mm hr⁻¹ per year (Fig. 3.1.6 in Kim 2020). Kim (2020) used the same TRMM 3B42 and definition of inner-core rainfall as in the current study, but the analysis period was different and TD was excluded. It would be good to compare the current study with Kim (2020).

Kim D. (2020). Rainfall structure of tropical cyclone over the globe. Ph.D. Diss. Seoul National Univ.

2. This study suggested increases in sea surface temperature and total precipitable water as the cause of the increase in TC rainfall. However, in general, TC rain increases as the TC gets stronger (Figure S1). Thus, long-term changes in TC rain can be associated with those in TC intensity. It seems necessary to analyze the change in TC intensity during the analysis period.

3. (Lines 132–144) Are the conditions unchanged in SIO? Are the changes in conditions significant for each ocean basin?

4. (Line 142) It is interesting that the RMR increases. RMR seems to be related to eyewall which usually appears inside the radius of maximum wind (RMW). In addition, the RMW tends to decrease as the TC gets stronger. This study found increases in sea surface temperature and total precipitable water which can increase TC intensity, so RMW and RMR may decrease. If the mechanism for the increase in RMR could be explained, it would be a very interesting finding.

5. (Line 27) Typo, "TC" would be "TCs".

6. (Figures 4e and 4f) Results for 1998 and 2016 are missing.

Reviewer #2 (Remarks to the Author):

Comments on "Global Increase in Tropical Cyclone Rain Rate".

This study provides the observational analysis in changes of tropical cyclone (TC) rain rate using investigating the Tropical Rainfall Measuring Mission and the Global Precipitation Measurement mission. The authors present that the global TC rain rate is increasing by 0.032 mm/h/year which accounts for 1.5% increase of TC rain rate every year. They also suggest that this increase in TC rain rate is combined result of decrease in the inner core rain rate and increase in the outer rainband rain rate. Changes in the sea surface temperature and total precipitation water support the changes in the TC rain rate. Overall, the manuscript is well written and novel, and the contents are very essential information in TC community. I enjoyed reading the manuscript and was

surprised to realize the absence of integrated observational investigation regarding the long-term change in TC rain rate. Yet, I have several major comments as written below.

Major comments:

1. The authors argue that the inner core (< 100 km) rain rate is decreasing but the outer rainband (200–500 km) rain rate is increasing. However, it is doubtful that the inner core rain rate is decreasing when considering the fact that the radius of maximum rain (RMR) is below 100 km in the earlier half (1998–2007) of the analysis period, while this RMR value exceeds over 100 km in the later half (2008–2016). Simple sensitivity test extending the inner core thresholds to 110 km, 120 km and etc. would be effective.

Besides, the authors need to provide the possible consequence after the omission of rain rate within the annular region between 100–200 km radii.

2. The authors suggest that the increase in the sea surface temperature and total precipitation water are relevant with the increase in the outer rainband rain rate. However, they lack of any explanation on why the inner-core rain rate is decreasing. Is it also related with warmer and moisture environment? Or is it possibly driven by other factors?

3. The proportion of inner-core rain rate and outer rainband rain rate constituting the total TC rain rate in climatology would be informative when understanding their offset.

4. Please provide some discussions on the results in relation with Lau and Zhou (2012) who analyzed the TC lifetime accumulate rain, which is decreasing in the Northeast and Northwest Pacific.

Minor question:

5. How did the authors calculate 5-year moving average for the first and last two years?

6. The punctuation mark in the unit “mm.h⁻¹/year” should be removed.

Reviewer #3 (Remarks to the Author):

Overall, this is a well-written and concise manuscript that explores how average precipitation has changed within tropical cyclones during the lifetime of TRMM. While, I think the headline result that rainfall rates within tropical cyclones has increased is important to the broader community, the methodology has some shortcomings:

1. I do not believe that the use of a hard-coded radius of 500 km is adequate. Yes, this has been standard in the community, but it could over estimate storm rainfall. A sensitivity analysis to this radius is a must.

2. I am not convinced that the breakdown between the inner-core and outer rainband regions is appropriate. This definition should be dynamic depending on the storm and the point within its lifetime.

3. Also, it seems like a mistake to only use TRMM every 6 hours to match the IBTrACS frequency. Is there a way to use all of the TRMM data?

I think that the above issues need to be addressed before this manuscript can be considered for publication in Nature Communications. In addition, I have provided some specific comments below (some of which are related to the above points).

Results:

L65–71: Note that while a radius of 500 km is standard in the community for such assessments, recent work by Stansfield et al. 2020 has suggested that such a “hard-coded” radius overestimates tropical cyclone related overland precipitation.

Stansfield, A. M., K. A. Reed, C. M. Zarzycki, P. A. Ullrich, and D. R. Chavas, 2020: Assessing Tropical Cyclones’ Contribution to Precipitation over the Eastern United States and Sensitivity to the Variable-Resolution Domain Extent. *J. Hydrometeor.*, 21, 1425–1445, doi: 10.1175/JHM-D-19-0240.1.

This is in part because the outer size (as well as RMW) of the storm can vary substantially across

basins, individual storms, and even within a storms lifetime.

Chavas, D. R., and Emanuel, K. A. (2010), A QuikSCAT climatology of tropical cyclone size, *Geophys. Res. Lett.*, 37, L18816, doi:10.1029/2010GL044558.

L88-89 (and L71-76): The paper really provides no background for why these definitions, particularly of the inner-core, are appropriate for this analysis. Especially considering the range of differences in observed storm RMW. Furthermore, Figure 4 proves that this estimate for the inner-core is variable and changes over time. Ultimately, the definition needs to be dynamical and vary from storm to storm.

L132-134: Is there a correlation in year-to-year variability between precipitable water or sea surface temperature and storm rainfall?

L135-144: What is the theory for this? Seems to contradict the work of:

Patricola, C.M., Wehner, M.F. (2018) Anthropogenic influences on major tropical cyclone events, *Nature*, 563, 339–346, doi: 10.1038/s41586-018-0673-2.

L198 and L214-215: How do the authors rectify the fact that the TRMM 3B42 and IBTrACS have different temporal frequencies? Do I interpret this correctly that only the 3 hourly precipitation every 6 hours is used for the analysis? This means that roughly half of the available TRMM precipitation is not used. Seems like a lost opportunity.

L218: What is the sensitivity to using 0.1 mm/hr. Do the results change significantly if this value is modified? Why not include all precipitating cells?

General responses to all reviewers (summary of all major changes):

We really appreciate all of your careful reviews and constructive comments. Based on the overall assessment of the reviewers and editor's suggestions, we have implemented the following major changes in the revision:

1. **Updated figures 1, 2, supplementary figure 1, and supplementary tables 2, 3, and 4:** We adjusted the definition of storm size from a fixed 500-km radius to a framework based on the concept of tropical cyclone precipitation features (TCPF). We included a sensitivity test in which both methods are compared. New figures 1 and 2 can be found in lines 470-479, updated Supplementary figure 1 in lines 24-26, and updated supplementary tables 2-4 in lines 14-30 of the supplementary material annex.
2. **Adjusted figure 3 and updated supplementary table 2:** We redefined the extent of the areas analyzed within the TC structure. In this revision, we examined the inner-core and the rainband region. This definition is different from the initial version because now we describe the full extent of the rainband instead of its outer section only. The new definitions are as follows:
 - a. Inner-core: From the TC center up to twice the radius of maximum rainfall. (2xRMR)
 - b. Rainband region: It is the area from 2xRMR until the external borders defined by the TCPF method.

New figure 3 can be found in lines 482–485 and the updated Supplementary table 2 in lines 14–16 of the supplementary materials annex.

3. **New supplementary figure 2:** A sensitive test comparing RMR, RMR x 1.5, and RMR x 2 is presented to describe the effects of varying the selection distances in estimating averaged inner-core rainfall rates.
New Supplementary figure 2 can be found in lines 33–38 of the supplementary materials annex. Discussion on the findings of this sensitivity test is in lines 117–128.
4. **New supplementary figure 3:** A new plot showing the percentual contributions of the inner-core and the rainband regions was created. This figure aims to help the readers to understand the variations in the rainfall rates better. Supplementary figure 3 can be found in lines 41–46 of the supplementary materials annex. Discussion on this plot can be found in lines 146–152.
5. **Adjustment of rainfall data samples, supplementary tables 6 and 7:** We switch from 6-hourly to 3-hourly best track positions to take advantage of the 3B42 temporal resolution. This adjustment did not change our findings of the global rainfall rates but slightly alter the slope of some of them. New rainfall rate trends have been updated through the document. Supplementary tables 6 and 7 can be found in lines 66–72 of the supplementary material annex.
6. **New discussion that compares Northern Hemisphere vs. Southern hemisphere:** Using the adjustments mentioned above, we found that SPA exhibit a slight trend, similar to SIO. Then, we adjusted the inter-basin comparison section to discuss the differences between Northern and Southern hemisphere basins. This discussion replaces our initial comparisons that addressed the

global differences with South Pacific only. New discussion can be found through the full Results and Discussion/conclusion sections, in lines 67 – 228 of the manuscript.

7. We also adjusted the numbering of figures in the supplementary materials annex using the order they are cited in the manuscript for the first time. Please use this table in case you want to track the changes.

New table or figure	Former table or figure
Supplementary table 1	Table S6
Supplementary table 2	Table S5 (Rain rate trends)
Supplementary table 3	Table S4
Supplementary table 4	Table S3
Supplementary table 5	Table S5 (TPW,SST)
Supplementary table 6	Table S1
Supplementary table 7	Table S2
Supplementary figure 1	Figure S1

Note: As mentioned above please notice this revision include additional new figures.

Finally, further analyses and discussion on our findings are provided to address the individual reviewer’s comments. Below you will see the specific responses (in black text) to each of their comments (in red text). Each answer also includes a reference to the new material, either to the line number or new figures and tables.

Answers to reviewer No.1

Comment No.1: *One of major findings in this study is the reduction in the inner-core rainfall rate. However, it seems that this result can be changed according to the definition of inner-core rainfall. In general, the inner-core means the area containing the eyewall where the strongest precipitation occurs. During the analysis period, the radius of maximum rain (RMR) increased from 95 to 105 km approximately. Therefore, the 100 km radius seems to be rather small to define the inner-core area. It seems that more accurate changes in inner-core rainfall can be found if the 120 or 150 km radius is used instead of the 100 km radius. In addition, similar results can be found in Kim (2020) which suggested that the inner-core rainfall rate decreases by 0.081 mm hr⁻¹ per year (Fig. 3.1.6 in Kim 2020). Kim (2020) used the same TRMM 3B42 and definition of inner-core rainfall as in the current study, but the analysis period was different, and TD was excluded. It would be good to compare the current study with Kim (2020). Kim D. (2020). Rainfall structure of tropical cyclone over the globe. Ph.D. Diss. Seoul National Univ.*

Answer:

We agree with the reviewer’s comment on the sensitivity of choosing the distance threshold that controls the inner-core extent. To address this issue, we have adjusted the definition in the text as follows: “In the case of the inner-core region, we used a storm-dependent method that uses the radius of maximum azimuthal rain rate (RMR) as the size determining factor. A sensitive test comparing RMR, 1.5*RMR, and 2*RMR was performed to describe the effects of varying the selection distances in estimating

averaged inner-core rainfall rates (Supplementary figure 2). Results indicate that the slopes of average rainfall rates slightly decrease as a longer radius is used to define the inner-core region. However, in all the inner-core definitions, the nearly same result is obtained". This adjustment can be found in lines 117-128.

About the RMR trend, due to changes made to use all 3B42 samples (now 3 hourly instead of 6 hourly before), we also updated our analysis as follows: "Considering that one of our observational results indicates that the inner-core rainfall rate is decreasing, further analysis of the RMR was performed. We found that globally and when considering all the TC categories, the RMR does not observe any particular trend (Supplementary figure 4a, 4b, and 4c). However, in CAT1 to CAT5 storms, results show that the RMR slightly expands outwards (Supplementary figures 4d, 4e, and 4f) with low R^2 and confidence levels (Supplementary table 2)." Please see lines 187-196 of the revised manuscript.

Regarding the opportunity to cross-check our findings with the research of Dr. Kim (2020), we have added the following paragraph: "In addition to the difference mentioned above, we found that rainfall rates at the inner-core level decrease for CAT1 to CAT5 hurricanes, which contradicts the results of modeling projections that suggest potential increases within 100 km of the TC center¹⁸. However, this finding is actually similar to the results of a recent observational study by Kim (2020)³⁶ which suggested that the inner-core rainfall rate decreases by 0.081 mm hr⁻¹ per year (his Fig. 3.1.6). Kim (2020)³⁶ used the same TRMM 3B42 data, but the analysis period was different and tropical depressions were excluded. The definition of inner-core rainfall was also different from this study. He used a fixed 100 km radius from the TC center to define inner-core rainfall. So far, we do not have a good explanation why the rain rate in the inner-core has a decreasing trend. This would be a direction for future studies." Please see lines 205-214 of the revised manuscript.

Comment No.2: This study suggested increases in sea surface temperature and total precipitable water as the cause of the increase in TC rainfall. However, in general, TC rain increases as the TC gets stronger (Figure S1). Thus, long-term changes in TC rain can be associated with those in TC intensity. It seems necessary to analyze the change in TC intensity during the analysis period.

Answer:

Thanks for the suggestion. To address this potential explanation, we included the following discussion in the new manuscript: "we also explored the change of maximum sustained speed during the analysis period (Supplementary table 5). However, statistical results show that storm intensities remain unchanged; therefore, we concluded that the increasing global trend must be the result of higher SST and TPW only. In other words, the scheme that assumes that the magnitude of rainfall is totally dependent on storm intensity is less than perfect because of the influence of environmental conditions." Please see lines 181-186 and Supplementary table 5 in lines 51-53 of the supplementary material annex.

Comment No.3: (Lines 132–144) Are the conditions unchanged in SIO? Are the changes in conditions significant for each ocean basin?

Answer:

Precipitation rates in the SIO ocean are also increasing (previous Figure 3e, now figure 2e in line 476), and the Kendall test indicates that the trend in this basin is statistically significant. However, compared to

the northern hemisphere trends, SIO slope is much flatter. In this revision, we decided to compare global trends not only against SPA but also SIO. Then a new discussion comparing northern and southern hemispheres is presented.

Comment No.4: *(Line 142) It is interesting that the RMR increases. RMR seems to be related to eyewall which usually appears inside the radius of maximum wind (RMW). In addition, the RMW tends to decrease as the TC gets stronger. This study found increases in sea surface temperature and total precipitable water which can increase TC intensity, so RMW and RMR may decrease. If the mechanism for the increase in RMR could be explained, it would be a very interesting finding.*

Answer:

Yes, our new findings show that RMR **only lightly** increases in global basins for storms from CAT1-CAT5. Based on the patterns shown in Figure 3 and Supplemental figure 4, we speculate RMR should experience a nearly 13-year cycle; however, it is difficult to conclude due to our time series' shortness. We are investigating further the increase of the RMR, which will be part of another publication.

Comment No.5: *(Line 27) Typo, "TC" would be "TCs"*

Answer:

Thanks! it has been adjusted as follows: "precipitation associated with tropical cyclones (TCs) has received more and more attention from the scientific community" please see lines 23-24 of the new manuscript.

Comment No.6: *(Figures 4e and 4f) Results for 1998 and 2016 are missing.*

Answer:

Thanks for noticing. These figures have been adjusted, including the missing data. In this revision, we split original figure 4 into two independent plots, figure 4 and supplemental figure 4. Please find these adjustments in lines 486-490 in the main manuscript, and lines 58-63 of the supplementary material annex.

Answers to reviewer No.2

Comment No.1: *The authors argue that the inner core (< 100 km) rain rate is decreasing but the outer rainband (200–500 km) rain rate is increasing. However, it is doubtful that the inner core rain rate is decreasing when considering the fact that the radius of maximum rain (RMR) is below 100 km in the earlier half (1998–2007) of the analysis period, while this RMR value exceeds over 100 km in the later half (2008–2016). Simple sensitivity test extending the inner core thresholds to 110 km, 120 km and etc. would be effective. Besides, the authors need to provide the possible consequence after the omission of rain rate within the annular region between 100–200 km radii.*

Answer:

We agree with the reviewer's comment on the sensitivity of choosing the distance threshold that controls the inner-core extent. To address this issue, we have adjusted the definition in the text as follows: "In the case of the inner-core region, we used a storm-dependent method that uses the radius of maximum azimuthal rain rate (RMR) as the size determining factor. A sensitive test comparing RMR, 1.5*RMR, and 2*RMR was performed to describe the effects of varying the selection distances in estimating averaged inner-core rainfall rates (Supplementary figure 2). Results indicate that the slopes of average rainfall rates slightly decrease as a longer radius is used to define the inner-core region. However, in all the inner-core definitions, the nearly same result is obtained". This adjustment can be found in lines 117-128.

About the comment on the 100-200 rainfall, in the new version we analyzed the entire rainband region instead of the outer rain bands.

Comment No.2: The authors suggest that the increase in the sea surface temperature and total precipitation water are relevant with the increase in the outer rainband rain rate. However, they lack of any explanation on why the inner-core rain rate is decreasing. Is it also related with warmer and moisture environment? Or is it possibly driven by other factors?

Answer:

Very good question! But we really don't know the answer for now. But it is a good future study direction to develop an explanation about the decrease in the inner-core region. We have added the following sentences to discuss this, "However, this finding is actually similar to the results of a recent observational study by Kim (2020)³⁶ which suggested that the inner-core rainfall rate decreases by 0.081 mm hr⁻¹ per year (his Fig. 3.1.6). Kim (2020)³⁶ used the same TRMM 3B42 data, but the analysis period was different and tropical depressions were excluded. The definition of inner-core rainfall was also different with this study. He used a fixed 100 km radius from the TC center to define inner-core rainfall. So far, we do not have a good explanation why the rain rate in the inner-core has a decreasing trend. This would be a direction for future studies.". Please see lines 207-214 of the revised manuscript.

Comment No.3: The proportion of inner-core rain rate and outer rainband rain rate constituting the total TC rain rate in climatology would be informative when understanding their offset.

Answer:

Thanks for the suggestion; we included the following description in the new version of the manuscript: "Despite the fact that the inner-core region produces the most intense rainfall rates (Supplementary figures 3a and 3b), the extent of the rainband region is much larger than the inner-core, which leads to a more significant contribution from the rainbands to the total TC rain (Supplementary figures 3c and 3d). These differences in the proportional contribution and the rising trend in the rainbands suggest that the global increase must be mainly a consequence of special conditions favoring rainfall production in the rainband environment." this adjustment can be found in lines 146-152.

Comment No.4: *Please provide some discussions on the results in relation with Lau and Zhou (2012) who analyzed the TC lifetime accumulate rain, which is decreasing in the Northeast and Northwest Pacific.*

Answer:

Thanks for the suggestion. We have added a few lines with comments regarding common arguments and discrepancies with their study, as follows: “An additional potential contradiction with our results is the findings of Lau and Zhou (2012)²⁰. They reported differences in the sign and magnitude of total TC lifetime accumulate rain trends between ATL (+23% per decade), ECPA (-25.1% per decade), and NWP (-20.9% per decade), which seemingly partially agrees with our findings in ATL but amply differed for ECPA and NWP. However, as mentioned in the introduction, Lau and Zhou (2012)²⁰ used the GPCP data which is a 5-day mean rainfall dataset from satellite and rain gauge measurements with a spatial resolution of 2.5° latitude X 2.5° longitude. With such a low temporal resolution, it was impossible to estimate realistic rain rate intensity in TCs. They recognized that the total TC lifetime accumulate rain measure they used was rather an integral measure of total rain energy associated with a TC. Therefore, their results are not comparable with our study here.” Please see lines 215-224 of the revised manuscript.

Comment No.5: *How did the authors calculate 5-year moving average for the first and last two years?.*

Answer:

Thanks for the suggestion. We have included further details about the procedure of calculating the 5-year moving average, including initial and final tails obtained from the MA smoothing model, as follows: “The 5-year moving average was calculated in two steps: 1) Values from 2000 to 2014 result from the traditional method. For instance, the first averaged value is obtained by taking the average of a 5-values subset (2 before, current, and 2 after) shifting forward until the end of the time series. 2) In the case of 1998, 1999, 2015, and 2016, we use the moving average model (MA Smoothing Model) to estimate the most suitable values. In the process, we tested several q-values and determined that q=2 is the best choice in all the cases. This value was selected by running the model over the entire series and then looking for the nearest series to the 5-year moving averages for 2000-2014. Finally, we included the modeled tails into the 5-year moving average to have the full coverage of the period.” This explanation can be found in lines 297-305.

Comment No.6: *The punctuation mark in the unit “mm.h⁻¹/year” should be removed.*

Answer:

Thanks for the suggestion. In the new version, we adjusted the unit to mmh⁻¹/year through the manuscript.

Answers to reviewer No.3

Comment No.1: *I do not believe that the use of a hard-coded radius of 500 km is adequate. Yes, this has been standard in the community, but it could overestimate storm rainfall. A sensitivity analysis to this radius is a must...*

Answer:

Thanks for the suggestion. The manuscript and calculations have been adjusted as follows: “Conventionally, the total TC rainfall area is defined by the 500 km radius threshold. This approach was used by several researchers in the previous studies^{21,22,23,24}. The rationale behind this distance threshold is explained by the findings of Englehart and Douglas (2001)²⁵, which indicates that in 90% of the TC cases, the distance between the center of the storm and the outer edge of its cloud shield is less than 550-600 km. However, we realize that the size of TCs can vary substantially across different basins, individual storms, and even within the lifetime of the same storm (Chavas and Emanuel 2010)²⁶. As suggested by Stansfield et al. (2020)²⁷, a “hard-coded” radius of 500 km could overestimate TC-related overland precipitation. Therefore, in this study, we define TC total raining area using a framework based on the concept of tropical cyclone precipitation features (TCPF, Jiang et al. 2011²²). In this TCPF framework, a precipitation feature (PF, or a rain cell) is defined by grouping contiguous pixels based on certain criteria. In this case, the criterion is 3B42 rain rate greater than 0.1 mm/h (see section 4b for more details). To be qualified as a TCPF, the distance between the TC center and the geometric center of the PF must be less than 500 km. We compared the traditional simple truncation of 500 km radius from the TC center (Figure 1a) and the TCPF-based TC total rain area definition (Figure 1b). Although both methods exhibit similar and consistent results, our findings focus on the TCPF approach since this provides a better estimation of the TC size²².” This adjustment can be found in lines 68-84.

Comment No.2: *I am not convinced that the breakdown between the inner-core and outer rainband regions is appropriate. This definition should be dynamic depending on the storm and the point within its lifetime.*

Answer:

We agree with the reviewer’s comment on the sensitivity of choosing the distance threshold that controls the inner-core extent and the rainbands. To address this issue, we have adjusted the definition in this research as follows: “In the case of the inner-core region, we used a storm-dependent method that uses the radius of maximum azimuthal rain rate (RMR) as the size determining factor. A sensitive test comparing RMR, 1.5*RMR, and 2*RMR was performed to describe the effects of varying the selection distances in estimating averaged inner-core rainfall rates (Supplementary figure 2). Results indicate that the slopes of average rainfall rates slightly decrease as a longer radius is used to define the inner-core region. However, in all the inner-core definitions, the nearly same result is obtained”. This adjustment can be found in lines 117-128. Additionally, we define the rainband region as the area between 2xRMR and the outermost extent of the storm.

Comment No.3: *Also, it seems like a mistake to only use TRMM every 6 hours to match the IBTrACS frequency. Is there a way to use all of the TRMM data?*

Answer:

Thanks for the suggestion. Our first version used 6-hour intervals because an essential part of our analyses is based on the SHIPS developmental database (6-hour). We wanted to use a homogeneous dataset for all the variables (i.e., paired observations), particularly during the statistical tests. However, to attend to the reviewers' concerns, we updated all the rainfall-related parameters using 3-hour intervals. This adjustment does not apply to the environmental variables, which remained 6-hourly. Please see Supplementary tables 6 and 7 in lines 66-72 of the supplementary material annex.

Comment No.4: L65-71: Note that while a radius of 500 km is standard in the community for such assessments, recent work by Stansfield et al. 2020 has suggested that such a "hard-coded" radius overestimates tropical cyclone related overland precipitation. Stansfield, A. M., K. A. Reed, C. M. Zarzycki, P. A. Ullrich, and D. R. Chavas, 2020: Assessing Tropical Cyclones' Contribution to Precipitation over the Eastern United States and Sensitivity to the Variable-Resolution Domain Extent. *J. Hydrometeor.*, 21, 1425–1445, doi: 10.1175/JHM-D-19-0240.1.

Comment No.5: This is in part because the outer size (as well as RMW) of the storm can vary substantially across basins, individual storms, and even within a storm's lifetime. Chavas, D. R., and Emanuel, K. A. (2010), A QuikSCAT climatology of tropical cyclone size, *Geophys. Res. Lett.*, 37, L18816, doi:10.1029/2010GL044558.

Answers to both comment #4 &5: Thanks for the comments and the reference papers. We totally agree with the reviewer's point of view. Both papers have been cited in the revised manuscript. Please see our answer to this issue in comment No.1. This adjustment can be found in lines 68-84.

Comment No.6: L88-89 (and L71-76): The paper really provides no background for why these definitions, particularly of the inner-core, are appropriate for this analysis. Especially considering the range of differences in observed storm RMW. Furthermore, Figure 4 proves that this estimate for the inner-core is variable and changes over time. Ultimately, the definition needs to be dynamical and vary from storm to storm.

Answer:

We agree with the reviewer's comment on the sensitivity of choosing the distance threshold that controls the inner-core extent and the rainbands. Please see our answer to this issue in comment No.2. This adjustment can be found in lines 117-128.

Comment No.7: L132-134: Is there a correlation in year-to-year variability between precipitable water or sea surface temperature and storm rainfall?

Answer:

Yes, we found a low positive correlation between precipitation rate and SST ($R=0.22$). In the case of total precipitable water, the correlation is moderate positive (0.50). Please see the figure below. We added a sentence in the revised manuscript: “We also found positive correlations between precipitation rate and SST with a correlation coefficient of 0.22, and between precipitation rate and TPW with a correlation coefficient of 0.50.” (lines 165-166).

Comment No.8: L135-144: What is the theory for this? Seems to contradict the work of: Patricola, C.M., Wehner, M.F. (2018) Anthropogenic influences on major tropical cyclone events, *Nature*, 563, 339–346, doi: 10.1038/s41586-018-0673-2.

Answer:

Thanks for pointing this out. Due to changes made to use all 3B42 samples (now 3 hourly instead of 6 hourly before) and the use of the TCPF method, our new findings show that RMR **only lightly** increases in global basins for storms from CAT1-CAT5. Therefore, we realized it is not reasonable to link the RMR trends with the inner-core trends. This whole paragraph has been removed in the revised manuscript.

Comment No.9: L198 and L214-215: How do the authors rectify the fact that the TRMM 3B42 and IBTrACS have different temporal frequencies? Do I interpret this correctly that only the 3 hourly precipitation every 6 hours is used for the analysis? This means that roughly half of the available TRMM precipitation is not used. Seems like a lost opportunity.

Answer:

Thanks for the suggestion. Please see our answer to this issue in comment No.2. Adjustments can be found in Supplementary tables 6 and 7 in lines 66-72 of the supplementary material annex.

Comment No.10: L218: What is the sensitivity to using 0.1 mm/hr. Do the results change significantly if this value is modified? Why not include all precipitating cells?

Answer:

Thanks for the question. We have added the explanation & sensitivity test descriptions as follows: “As mentioned above, in this study, we use storm-dependent definitions to define the total TC raining area and the inner-core raining area. In the TCPF framework to define the total TC raining area, a minimum value of precipitation rate must be used to calculate each PF or rain cell. Ideally, this threshold should be rain rate = 0 mm/h. However, the infrared or microwave rainfall retrievals in TRMM/GPM 3B42 are often contaminated by non-raining signals, especially for light rains in the retrievals. This contamination could cause problems in our TCPF identification. Therefore, we tested a few minimum values as the threshold to define rain cells, including 0.01, 0.1, and 0.25 mm/h. The sensitivity test confirms that the rising global trend still presents in all scenarios. The intermediate value of 0.1mm/h is chosen in this study.” This adjustment can be found in lines 265-274.

Second round review --

Reviewer #1 (Remarks to the Author):

The author has responded appropriately to all questions and revised the manuscript. Therefore, it is recommendable to accept this paper. Several typos have been found, so please correct them in proofreading.

1. The results of Kim et al. (2021) may support the conclusion of this study. Kim et al. (2021) suggested that "Inner-core rainfall strength (RS) is strongly correlated with Vmax, but weakly correlated with large-scale environmental conditions. In contrast, total rainfall area (RA) showed a stronger correlation with environmental conditions than with Vmax (Summary and discussion part)." The RA corresponds to the total TC raining area which is closely related to the rainband environment in this study. In other words, TC rainfall outside the core is more sensitive to environmental conditions than that inside the core, which may explain the finding that only rainfall in the rainband increased with increasing SST and TPW.

Kim, D., C.-H. Ho, H. Murakami, and D.-S. R. Park, Assessing the Influence of Large-Scale Environmental Conditions on the Rainfall Structure of Atlantic Tropical Cyclones: An Observational Study. *Journal of Climate*, 34(6), 2093-2106, (2021).

2. (Line 100) It looks like there should be "are" between "Pacific" and "near".

3. (Line 199) Please put a space between "expected" and "(modeled)".

4. (Line 238) Please put a space between "3B42" and "(version 7)".

Reviewer #3 (Remarks to the Author):

Again, this is a well-written and concise manuscript that explores how average precipitation has changed within tropical cyclones during the lifetime of TRMM. The revised version of the manuscript has addressed my main comments adequately. I think the manuscript is now suitable for publication in *Nature Communications* once a few minor comments are addressed:

L61: change "mainly" to "can"

L111: I'm confused by "near to 0.03." Table S4 suggests that they range from 0.015 to 0.043. I suggest changing to "in the range of 0.015 to 0.043," or something like that.

L120: It seems (or maybe I missed it) that including definition of the inner-core as a simple truncation of 100 km in Supplementary Figure 2a is not discussed in the text.

L126: Remind readers of what the external borders of TCPF are here (500 km?).

L140: Specify the time period here (i.e., move up from L142-143) so that the implications of this statement are taken out of context.

General responses to all reviewers (summary of changes):

We really appreciate all your careful reviews and constructive comments. Based on the overall assessment of the reviewers and editor's suggestions, we have implemented the following minor changes in the final revision:

1. We extended the discussion and conclusions section to include additional supporting evidence provided by a recent study from Kim et al. (2021). This adjustment can be found in lines 189 to 196.
2. We adjusted all the writing and redaction suggestions provided by the reviewers and the editor.

Please find below the specific responses (in black text) to each of the reviewers' comments (in red text). Each answer also includes a reference to the new line numbers in the improved manuscript.

Answers to reviewer No.1

Comment No.1: *The results of Kim et al. (2021) may support the conclusion of this study. Kim et al. (2021) suggested that “Inner-core rainfall strength (RS) is strongly correlated with Vmax, but weakly correlated with large-scale environmental conditions. In contrast, total rainfall area (RA) showed a stronger correlation with environmental conditions than with Vmax (Summary and discussion part).” The RA corresponds to the total TC raining area which is closely related to the rainband environment in this study. In other words, TC rainfall outside the core is more sensitive to environmental conditions than that inside the core, which may explain the finding that only rainfall in the rainband increased with increasing SST and TPW.*

Kim, D., C.-H. Ho, H. Murakami, and D.-S. R. Park, Assessing the Influence of Large-Scale Environmental Conditions on the Rainfall Structure of Atlantic Tropical Cyclones: An Observational Study. Journal of Climate, 34(6), 2093-2106, (2021).

Answer:

Thanks for the suggestion! The manuscript has been adjusted to include this important reference as follows: “We found that the results of Kim et al. (2021) 35 may partially support our findings. Their study suggest that the strength of inner-core rainfall strength is strongly correlated with TC intensity, but weakly correlated with large-scale environmental conditions, while the TC total rainfall area showed a stronger correlation with environmental conditions than with TC intensity. The total TC raining area is closely related to the rainband environment. In other words, TC rainfall outside the core is more sensitive to environmental conditions than that inside the core, which may explain the finding that only rainfall in the rainband increased with increasing SST and TPW.” please see lines 189-196 of the updated manuscript.

Comment No.2: *(Line 100) It looks like there should be “are” between “Pacific” and “near”.*

Answer:

Thanks! it has been adjusted as follows: “Rising trends across the Northern Indian Ocean are centered around 0.03 mmh-1 /year (Figure 2d), and trends in the East and Central Pacific **are** near 0.018mmh-1 /year (Figure 2b).” please see lines 100-101 of the new manuscript.

Comment No.3: *(Line 199) Please put a space between “expected” and “(modeled)”.*

Answer:

Thanks for the suggestion! it has been adjusted as follows: “However, the differences in the order of magnitude between the expected_(modeled) vs. observed rainfall rates in our period of study are substantial” please see lines 208-209 of the updated manuscript.

Comment No.4: *(Line 238) Please put a space between “3B42” and “(version 7)”.*

Answer:

Thanks for the suggestion! it has been adjusted as follows: “Rainfall information is obtained from the multi-sensor precipitation estimate TRMM 3B42_(version 7) for the period January 1998 to September 2014, and the transitional product that uses the new GPM constellation for the period September 2014 to December 2016” please see lines 247-248 of the updated manuscript.

Answers to reviewer No.3

Comment No.1: *L61: change “mainly” to “can”*

Answer:

Thanks for the suggestion! it has been adjusted as follows: “Analysis of environmental parameters shows that these trends are associated with increases in sea surface temperature and total precipitable water that **can** affect the TC rainband environment.” please see lines 61-62 of the updated manuscript.

Comment No.2: *L111: I am confused by “near to 0.03.” Table S4 suggests that they range from 0.015 to 0.043. I suggest changing to “in the range of 0.015 to 0.043,” or something like that..*

Answer:

it has been adjusted as follows: “In all the categories, there is a consistent increase **in the range of 0.015 to 0.043 mmh-1/year.**” please see lines 111-112 of the updated manuscript.

Comment No.3: L120: *It seems (or maybe I missed it) that including definition of the inner-core as a simple truncation of 100 km in Supplementary Figure 2a is not discussed in the text.*

Answer:

Thanks for noticing it! It has been adjusted as follows: “A sensitive test comparing a simple truncation of 100 km, RMR, 1.5xRMR, and 2xRMR was performed to describe the effects of varying the selection distances in estimating averaged inner-core rainfall rates.” please see lines 119-120 of the updated manuscript.

Comment No.4: L126: *Remind readers of what the external borders of TCPF are here (500 km?).*

Answer:

Thanks for the suggestion! it has been adjusted as follows: “Finally, for the rainband region, we used the area from 2xRMR until the external borders defined by the TCPF method that includes precipitating features with centroids within 500 km from the TC center; this selection was made as an approximation to allow the inclusion of rainfall rates in the area beyond two times the radius of maximum wind and the outermost part of the storm.” please see lines 126-130 of the updated manuscript.

Comment No.6: L140: *Specify the time period here (i.e., move up from L142-143) so that the implications of this statement are taken out of context.*

Answer:

it has been adjusted as follows: “In this study, an increasing trend in the TC rainfall rate during the period 1998-2016 has been identified on a global scale. On average, an increase of 0.027mmh⁻¹ /year is observed in the time series.” please see lines 142-143 of the updated manuscript.